# On the Use of NDVI to Estimate LAI in Field Crops: Implementing a Conversion Equation Library

**Sofia Bajocco** [1], **Fabrizio Ginaldi** [2,*], **Francesco Savian** [2], **Danilo Morelli** [1], **Massimo Scaglione** [1], **Davide Fanchini** [2], **Elisabetta Raparelli** [1] and **Simone Ugo Maria Bregaglio** [2]

[1] Council for Agricultural Research and Economics, Research Centre for Agriculture and Environment (CREA-AA), Via della Navicella 2-4, 00184 Rome, Italy; sofia.bajocco@crea.gov.it (S.B.); danilo.morelli@crea.gov.it (D.M.); massimo.scaglione@crea.gov.it (M.S.); elisabetta.raparelli@crea.gov.it (E.R.)

[2] Council for Agricultural Research and Economics, Research Centre for Agriculture and Environment (CREA-AA), Via di Corticella 133, 40128 Bologna, Italy; francesco.savian@crea.gov.it (F.S.); davide.fanchini@crea.gov.it (D.F.); simoneugomaria.bregaglio@crea.gov.it (S.U.M.B.)

\* Correspondence: fabrizio.ginaldi@crea.gov.it

**Abstract:** The leaf area index (LAI) is a direct indicator of vegetation activity, and its relationship with the normalized difference vegetation index (NDVI) has been investigated in many research studies. Remote sensing makes available NDVI data over large areas, and researchers developed specific equations to derive the LAI from the NDVI, using empirical relationships grounded in field data collection. We conducted a literature search using "NDVI" AND "LAI" AND "crop" as the search string, focusing on the period 2017–2021. We reviewed the available equations to convert the NDVI into the LAI, aiming at (i) exploring the fields of application of an NDVI-based LAI, (ii) characterizing the mathematical relationships between the NDVI and LAI in the available equations, (iii) creating a software library with the retrieved methods, and (iv) releasing a publicly available software as a service, implementing these equations to foster their reuse by third parties. The literature search yielded 92 articles since 2017, where 139 equations were proposed. We analyzed the mathematical form of both the single equations and ensembles of the NDVI to LAI conversion methods, specific for crop, sensor, and biome. The characterization of the functions highlighted two main constraints when developing an NDVI-LAI conversion function: environmental conditions (i.e., water and light resource, land cover, and climate) and the availability of recurring data during the growing season. We found that the trend of an NDVI-LAI function is usually driven by the ecosystem water availability for the crop rather than by the crop type itself, as well as by the data availability; the data should be adequate in terms of the sample size and temporal resolution for reliably representing the phenomenon under investigation. Our study demonstrated that the choice of the NDVI-LAI equation (or ensemble of equations) should be driven by the trade-off between the scale of the investigation and data availability. The implementation of an extensible and reusable software library publicly queryable via API represents a valid mean to assist researchers in choosing the most suitable equations to perform an NDVI-LAI conversion.

**Keywords:** field crop; leaf area index; normalized difference vegetation index; remote sensing; SaaS; software component

## 1. Introduction

The leaf area index (LAI) is defined as the ratio of the one-sided leaf surface area per unit ground surface area [1]. The LAI is directly related to the crop growth dynamics, the geometry of the vegetation canopy, as well as to ecological processes at a global and regional scale [2]. The LAI is often used in modeling biophysical processes and represents a key input for crop growth estimation and yield forecasting activities [3,4]. The reliable assessment of the LAI is therefore a major concern; however, many related factors are

far from being trivial, such as the complexity of the canopy architecture, the internal arrangement of leaves, and the background effects of the soil [5].

The LAI can be either directly measured by destructive leaves sampling or indirectly assessed using devices such as a ceptometer or LAI-2000 [6]. Both approaches are labor-intensive, time-consuming, and expensive [7,8]. Furthermore, although these in situ techniques can be accurate, they are not practical for the spatially explicit and continuous monitoring of an LAI over large geographic areas [9]. Remote sensing has then been widely used to fill these gaps by delivering timely LAI data. Advances in remote sensing infrastructure have led to relevant improvements in mapping and monitoring crop dynamics, as well as in crop yield forecasting [10]. The advantage of using a remote sensing technique is its applicability at multiple scales, according to the study objective and the sensor used: from global applications for agro-ecological and agro-climatic studies, to the sub-field scale for precision farming [5].

There is a strong interest in developing methodologies for the remote estimation of the LAI [11,12], to be used as an indicator of crop vegetation status [13] or to be assimilated into crop growth models [14,15]. Several remote sensing techniques for LAI estimation have been explored at different spatial scales and over different canopy structures [2,3,16]. Empirical methods are one of the most widespread techniques to estimate the LAI, and they mainly consist of exploiting the correlations between the LAI and some vegetation indices (VIs). VIs are widely used in remote sensing, primarily due to their easy derivation and applicability [11]. They are specific combinations of various spectral bands which allow for evaluating the plant status from images. Because the vegetation shows a strong absorption in the red spectral range (depending on plant chlorophyll) and a high reflectance in the near-infrared bandwidth (depending on the intercellular structure of the leaves mesophyll) [17], VIs combining these spectral responses may provide an indicator of vegetation "greenness", and hence a proxy of the LAI and chlorophyll content [18]. Accordingly, a simple ratio (SR) [19], normalized difference vegetation index (NDVI) [20], and soil-adjusted vegetation index (SAVI) [21] are among the most used VIs to estimate the LAI [7].

The NDVI is, by far, the most used and stable VI for estimating the LAI [5]. Nevertheless, its relationship with the LAI is essentially non-linear [22], showing high sensitivity to changes in the crop canopy at early growth stages (low LAI) and saturating when the crop canopy becomes dense [2,9]. New approaches have been proposed to overcome this limit, by using, for instance, spectral regions in the green and red edge [23–25]. However, data from the red-edge spectral region are not always available, and the green band is usually available at a coarser spatial resolution than other bands [22]; consequently, the derivation of the LAI from the NDVI is still widely applied in the literature.

The soundness of the empirical equation correlating the NDVI and LAI depends on the variability and quality of the data in the specific conditions tested [8]. Many research studies investigated the crop-specific relationship between the LAI and NDVI at different times, sites, and biomes [8,26–28]. However, the robustness and the transferability of empirical LAI-NDVI relationships to other regions may potentially be altered by many factors, such as sun-surface sensor geometry, crop management practices, and environmental and climatic conditions [7]. Moreover, the canopy cover reflectance depends on multiple variables related to crop seasonality and distribution patterns [29,30]. Consequently, a plethora of NDVI-based LAI equations have been proposed and published in the literature, deriving from applications in different regions, on different crops, and with different remote sensors.

Shedding light on the use of the NDVI to estimate the crop LAI is then a key field of investigation to unravel the different methodological approaches and practical aspects covered by the equations proposed in the literature so far. The objective of the work is therefore to perform a quantitative assessment of the state-of-the-art of the methodologies used to estimate the LAI from the NDVI in agriculture, by (i) exploring the use of an NDVI-based LAI in the literature, (ii) characterizing the NDVI-LAI algorithms available in the literature in terms of crop, type of equation, sensor, and biome, (iii) creating a library with

the NDVI-LAI algorithms available in the literature, and (iv) releasing a publicly available software as a service (SaaS) to integrate remote sensing data with crop simulation models.

## 2. Materials and Methods

### 2.1. Collecting and Characterizing the NDVI-LAI Equations

We conducted a literature search on agriculture and crops using Google Scholar database, focusing on articles, conference proceedings, reviews, book chapters, notes, articles in press and letters, published in English language. We used "NDVI" AND "LAI" AND "crop" as search string, focusing on the period 2017–2021. Only the publications where the conversion equation was reported were retained in the analysis.

The crops under investigations were wheat, maize, barley, rice, vineyard, soybean, esunflower, sugarcane, pasture, poplar plantations, and mixed land cover.

We extracted the NDVI-LAI conversion equation and categorized the mathematical form as: linear, exponential, power, polynomial, and logarithmic; when present, we also reported the coefficient of determination ($R^2$) as accuracy index.

The sensors used to derive NDVI values were characterized (i.e., field, airborne, or spaceborne), and we grouped them in four categories according to their spatial resolution: very high (<1 m), high (1–10 m), moderate (10–30 m), and low (>30 m).

Finally, we extracted the geographic coordinates (latitude and longitude) of the experimental field where the study was executed. For those articles where the coordinates were not clearly stated, we referred to the geographic location of the region, city, or county cited in the text. A bioclimatic attribute was assigned to each NDVI-LAI equation, by intersecting the equation dataset with the Ecoregion map of the world (https://ecoregions.appspot.com, accessed on 23 May 2022 [31]) using a Geographic Information System (GIS) environment to derive the biome category, i.e., ecosystems with similar climate, topography, and soils, and characterized by distinctive association of plants and animals [32]. We grouped the biomes identified into: Xeric, Mediterranean, Temperate, and Tropical.

The resulting dataset entailed the following attributes for each NDVI-LAI equation: (i) the equation type, (ii) the coefficient of determination, (iii) the sensor, (iv) the spatial resolution of the sensor, (v) the crop, and (vi) the biome of the experimental field (https://doi.org/10.6084/m9.figshare.20359437.v2). When the same equation was used in the same publication for different crops, sensors, or biomes, we replicated the corresponding record with all relevant attributes. To characterize each equation, we considered the positive part of the whole NDVI range of existence (i.e., from 0 to 1) and computed all the possible corresponding LAI values according to the different NDVI-LAI conversion methods identified. As for NDVI saturation, when LAI values were not upper limited (e.g., logarithmic equations), the maximum LAI value was assigned equal to the maximum value reported in the corresponding publication. The results of the different equations have been analyzed by grouping the NDVI-LAI values per crop, equation type, sensor spatial resolution, and biome and exploring the corresponding confidence intervals together with the distribution of the $R^2$ values declared in the articles.

### 2.2. Developing a Software Library Implementing NDVI-LAI Equations

Conversion equations retrieved from literature were implemented in a software component which provides a structured repository of methods to estimate crop LAI from NDVI values. NDVI-LAI equations were categorized by the sensor they were derived from, the crop they refer to, and the biome of the experimental field.

The equation library was implemented in a software component written as C# libraries and compiled for the Windows NET 4.6.2 framework. Software design follows the guidelines of the BioMA modeling platform [33], which aims at encapsulating modeling problems into discrete, specific domain, and reusable software units (components). Within this framework, the definition of the input/output (I/O) data structures is separated from the modeling approaches which use them, according to an implementation of the Bridge

pattern [34], in order to let developers to further extend available algorithms without any change in data structures.

Equations are implemented as simple strategies (https://doi.org/10.6084/m9.figshare.20359437.v2), which are units of code (C# classes) isolating a single algorithm for the conversion of a specific proxy into state/rate variables of the cropping system (e.g., satellite-derived NDVI into LAI). Composite strategies allow to compose simple strategies into higher-level procedures constituted by a sequential call of multiple strategies.

The component also includes methods to perform quality check of input (pre-conditions) and output (post-conditions) variables and equation parameters, according to their ontology (minimum, maximum, and default value, unit, type, and description).

The algorithms included in the library are fully documented at https://doi.org/10.6084/m9.figshare.20359437.v2.

### 2.3. Release of Software as a Service (SaaS)

NDVI-LAI conversion functions are served as RESTful APIs, a well-established software architectural style among services offered on the internet, and an open standard in data transmission and publication in the Cloud. API call enables users to query the model component either by single functions or multiple function attributes (crop, sensor, biome) via web, obtaining in the latter case a set of estimations as a result.

The RESTful protocol is developed on top of the HTTP protocol. HTTP calls and responses are manageable, with well-established libraries, in virtually every technological stack, e.g., Java, Python, C#, R. The data exchange format chosen is JSON.

The BioMA component has been adapted to run in an SaaS architecture, relying on Microsoft Azure as Cloud services provider, and any piece of elaboration is executed by an instance of an Azure Function, a stateless unit of elaboration. Therefore, the use of the service does not require any on-premises installation by models' user, as the business logic is contained and executed on the server side. Consequently, updates to the business logic do not require any further installations and, in case of updates to the business logic, no new documentation is to be sent to users unless the invocation interface changes. Furthermore, no functional dependencies need to be installed on the invoking client side, apart from libraries to manage HTTP calls and to parse and serialize JSON calls and responses payloads. Finally, third party access can be configured on the server side, allowing calls to be complemented with an access token that can be granted to a party having an agreement with the APIs' publisher. The documentation of the RESTful API call is reported in the Supplementary Material File S1.

### 2.4. Application Examples

To test the applicability of the developed SaaS, we used wheat and maize as case studies, selected as representative of fall–winter and summer crops. We focused on selected study plots where we are confident about the crop cultivated to avoid spurious pixels and signal noises, while ensuring a meaningful spatial heterogeneity.

We referred to the maize and wheat cadastral maps of 2018 provided by the Italian National Statistics Institute (ISTAT) and selected two plots where the field coverage was largely homogeneous: wheat in Southern Italy (Foggia, Mediterranean biome) and maize in Northern Italy (Vicenza, Temperate biome). Then, we referred to the freely available Sentinel 2 satellite data because of the high spatial resolution, to minimize the probability of having mixed pixels. So, for each plot, we downloaded the Sentinel 2 red and NIR data for the year 2018 and computed the NDVI for the available images. To reduce the noise and have a regular time-series, we computed the NDVI monthly maximum value composite (MVC) [35], obtaining 12 NDVI images per year per pixel. Pixel values were then averaged to obtain the mean annual NDVI profile of each plot. NDVI values were converted into LAI using the SaaS and selecting equations based on three different use scenarios (Table 1) defined as the intersection between: (i) crop- and biome-specific equations, regardless of the spatial resolution; (ii) crop- and spatial resolution-specific equations, regardless of

the biome; (iii) biome- and spatial resolution-specific equations, regardless of the crop. As an example, for maize we selected: maize-specific and Temperate equations, maize-specific and high-resolution equations, and Temperate and high-resolution equations. To the contrary, for wheat, we selected: wheat-specific and Mediterranean equations, wheat-specific and high-resolution equations, and Mediterranean and high-resolution equations. Finally, the resulting LAI seasonal profiles were analyzed according to each case study and use scenario.

**Table 1.** List of the attributes of the two case studies and corresponding use scenarios. ∩ indicates the intersection between two specific attributes of the case studies.

| Case Study Attributes | | |
|---|---|---|
| **Features** | **Case Study 1** | **Case Study 2** |
| Crop | Maize | Wheat |
| Biome | Temperate | Mediterranean |
| Sensor | Sentinel 2 | Sentinel 2 |
| **Use scenarios (queries submitted to SaaS service)** | | |
| Crop ∩ Biome | Maize ∩ Temperate | Wheat ∩ Mediterranean |
| Crop ∩ Spatial resolution | Maize ∩ High | Wheat ∩ High |
| Biome ∩ Spatial resolution | Temperate ∩ High | Mediterranean ∩ High |

## 3. Results

The literature search provided 92 articles from which 139 equations were extracted. The number of equations for the different crops was 65 for wheat, 57 for maize, 17 for rice, 9 for barley, 6 for vineyard, 4 for sugarcane and pasture, 3 for soybean, 2 for sunflower and mixed land cover, and 1 for poplar plantations. The characterization of the dataset as well as the corresponding NDVI-LAI equations library are available at https://doi.org/10.6084/m9 .figshare.20359437.v2. For the sake of simplicity, in this work, only the results for maize and wheat, which are the two most widely grown staple crops worldwide [36], are presented.

For all the equations identified, exponential and linear were the most frequent mathematical relationships proposed to convert the NDVI to the LAI in both crops (Figure 1). While on maize the use of two forms was balanced (34% linear and 32% exponential), the exponential form was largely the most used on wheat (47%). The exponential form actually represents the most widely accepted relationship between the NDVI and the LAI, being characterized by NDVI saturation at dense canopy cover; yet, linear equations are still used when few experimental LAI values are collected, reflecting a specific sampling moment rather than the whole growing season.

The Temperate biome was the most frequent on both maize (67%) and wheat (57%), followed by the Tropical (19%) on maize, and the Mediterranean (16%) and Xeric (16%) on wheat (Figure 1). This distribution well reflects the environmental suitability of the two crops: maize is a C4 plant with a high productivity under well-watered conditions, while wheat is a winter rainfed cereal also adapted to semi-arid environments.

According to Figure 1, the mostly used sensors to derive the NDVI data on maize were moderate-resolution satellites (41%), like Landsat, and very high spatial resolution sensors (25%), like field-based or airborne instruments. To the contrary, on wheat, we found the prevalence of field and drone cameras (37%), followed by high- (25%) and moderate-resolution (23%) satellites. The larger dimension of the archive, the free image availability, and the higher temporal resolution may explain the predominance of the moderate-resolution satellites with respect to those with very high and high spatial detail. Furthermore, satellites with 30 m pixel size may still be suitable for crops with homogeneous and large field coverage, like maize and wheat.

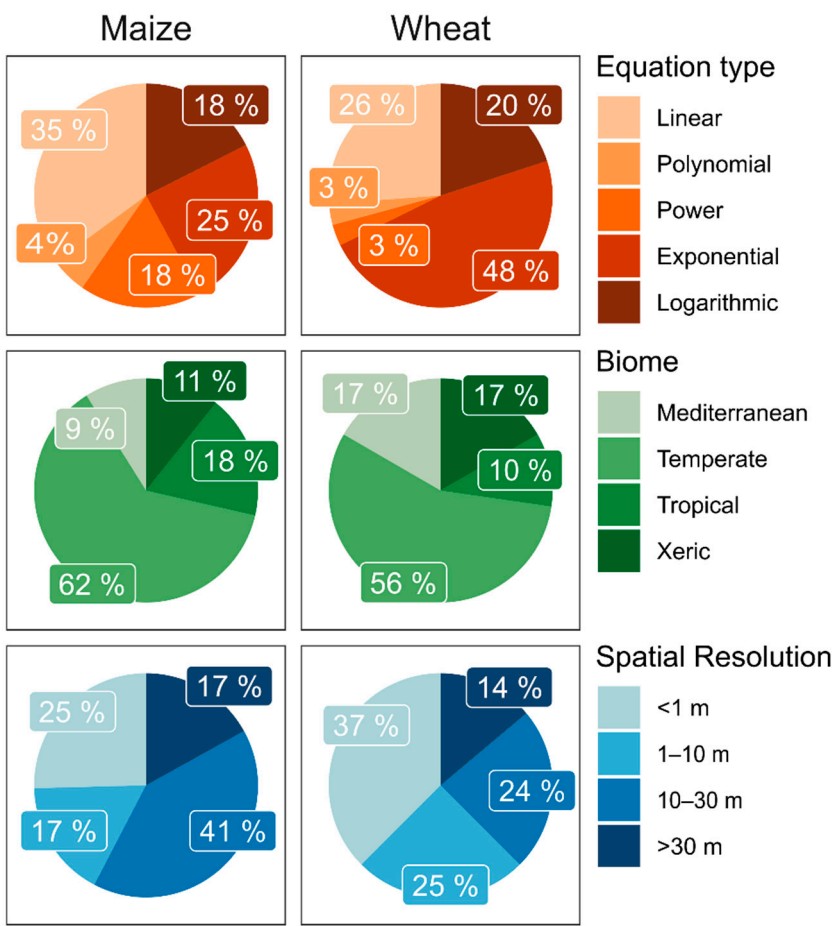

**Figure 1.** Pie charts describing the percentage of equations for maize and wheat sorted by (**top**) mathematical form (total number of collected equations for maize, *n* = 57; *n* = 65 for wheat), (**middle**) biome (*n* = 57 for maize; *n* = 66 for wheat), and (**bottom**) sensor spatial resolution (*n* = 60 for maize; *n* = 72 for wheat).

The characterization of the NDVI-LAI conversion equations was made in terms of the sensor spatial resolution, biome, and equation type, and the corresponding computation of the LAI values starting from the NDVI positive range of existence (i.e., from 0 to 1) has provided the results shown in Figures 2–4.

On maize, most of the NDVI-LAI conversion equations had a linear relationship (n = 20), developed in the Temperate zones (n = 35), and used moderate-resolution satellite images (n = 24). On the contrary, on wheat, the equations mainly used an exponential form (n = 31) and were mainly applied in Temperate biomes (n = 37) using field or airborne data (n = 27).

On both crops, the narrowest 25–75th percentile function distribution area corresponded to the exponential form, for which the highest $R^2$ values ($R^2$ > 0.5) have been also reported (Figure 2). This highlights that, in the selected studies, the exponential form is more accurate with respect to other forms, like, e.g., the linear relationship. In this latter case, several inconsistencies emerged, e.g., a largest divergence among different equations, an $R^2$ lower than 0.5, and an LAI = 0 when the NDVI < 0.3. The linear equation form is generally used when only a few spot data during the season are available and is then less representative of the actual non-linear relationship between the NDVI and LAI during the season. It is noteworthy that all the function ensembles, regardless of the equation type, tend to follow an exponential dynamic with a saturation around the LAI values equal to 5.

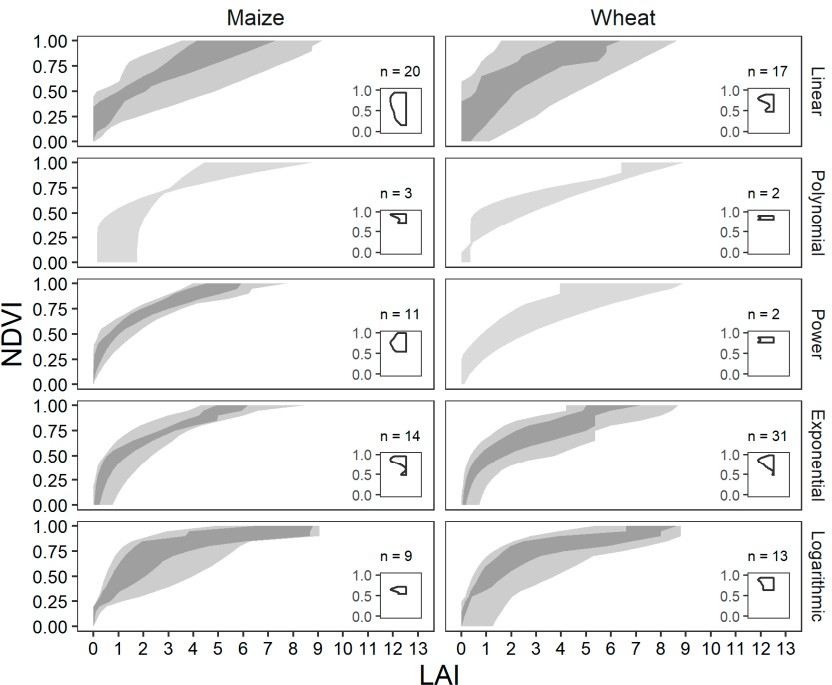

**Figure 2.** Correlation plots between the positive part of the normalized difference vegetation index (NDVI) existence range and leaf area index (LAI) for maize and wheat, sorted by mathematical form. The grey shadows indicate the 5–95th (light grey) and the 25–75th percentile (dark grey) of the LAI values obtained from the equations considered. The half-violin plot shows the $R^2$ distribution; n is the number of equations.

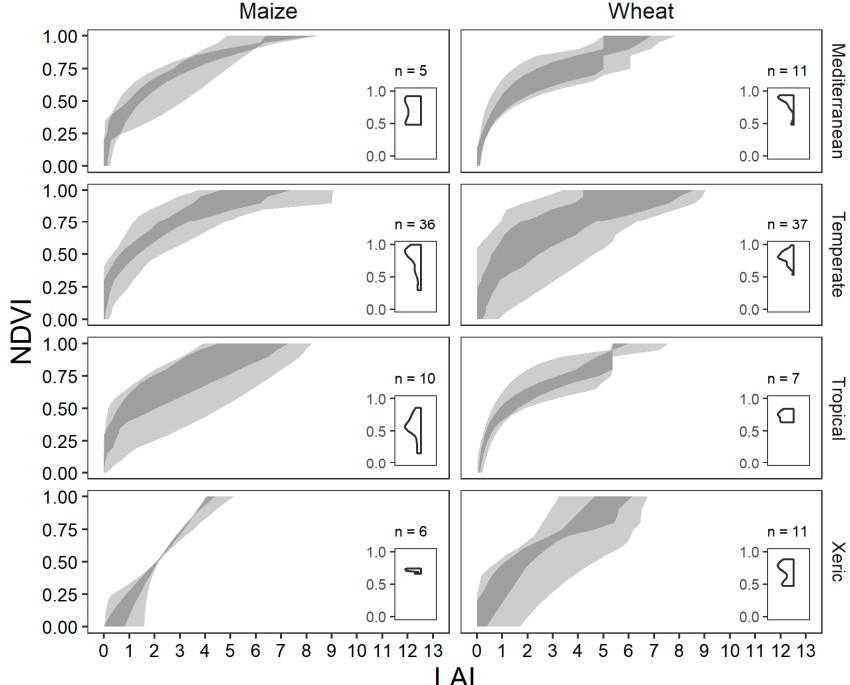

**Figure 3.** Correlation plots between the positive part of the normalized difference vegetation index (NDVI) existence range and leaf area index (LAI) for maize and wheat, sorted by biome. The grey shadows indicate the 5–95th (light grey) and the 25–75th percentile (dark grey) of the LAI values obtained from the equations considered. The half-violin plot shows the $R^2$ distribution; n is the number of equations.

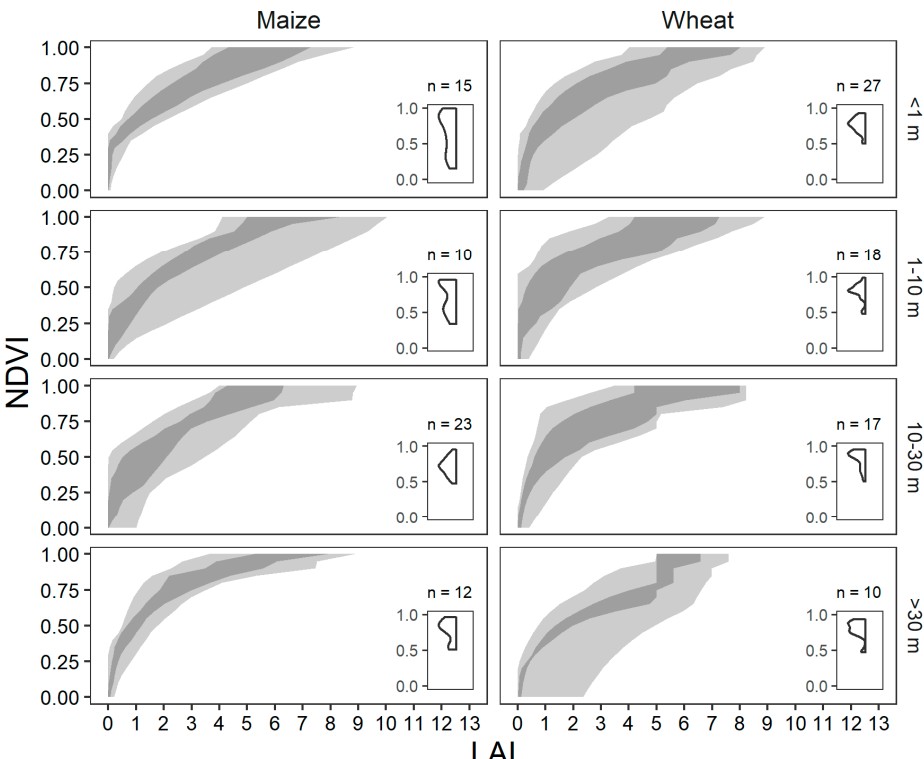

**Figure 4.** Correlation plots between the positive part of the normalized difference vegetation index (NDVI) existence range and leaf area index (LAI) for maize and wheat, sorted by sensor spatial resolution. The grey shadows indicate the 5–95 (light grey) and the 25–75th percentile (dark grey) of the LAI values obtained from the equations considered. The half-violin plot shows the $R^2$ distribution; n is the number of equations.

As for the biomes where the NDVI-LAI equations have been developed, both for maize and wheat, the Temperate conditions were those mostly studied (Figure 3). However, the highest $R^2$ values ($R^2 > 0.5$) have been recorded in the Mediterranean regions, where the narrowest 25–75th percentile area was also observed, suggesting that the NDVI-LAI algorithms developed for the Mediterranean biomes are mutually more consistent with respect to other regions. To the contrary, the largest divergence among the different equations was detected in the Temperate biomes for wheat and the Tropical biomes for maize, where, in addition, most equations presented values of the LAI equal to zero with NDVI values up to 0.3. These bioclimatic conditions are those with the highest variability thanks to the presence of a wide spectrum of environments from dry to wet; thus, the functions developed tend to each cover a different range of environmental conditions and have distinct behaviours.

The 25–75th percentile of the equation distribution area was narrower for low spatial resolution sensors on maize, indicating a higher consistency among the corresponding equations (Figure 4). The same result was obtained on wheat, where high and moderate resolutions showed a larger width of the 25–75th percentile distribution. It is to be noticed that most equations with 1–10 m spatial resolution on wheat and 10–30 m on maize presented an LAI = 0 when the NDVI < 0.4. The LAI tended to saturate at values around 5 on both crops and considering all spatial resolutions. The best accuracy was associated to the NDVI-LAI equations derived from low-resolution images ($R^2 > 0.5$) on both crops. Our results highlighted a higher consistency in the LAI estimation from equations using low-resolution satellites rather than very high resolution data, on both crops; this could be due, on one side, to the higher spatial heterogeneity detectable with fine-scale observations, and, on the other side, to structural errors sensor-dependent associated to human-based scanning, detection time, flight conditions, etc. These issues tend to be overcome with

satellite-based observations that flatten intra-pixel heterogeneity and ensure recording consistency and noise-removal in the pre-processing phase.

Examples of the application of the RESTful API to perform the NDVI-LAI conversion are presented in Figure 5; the LAI seasonal dynamics derived from all the equations using the MVC monthly NDVI data as input are reported for the maize and wheat plots. The LAI maximum values are in line with the literature data for wheat grown in Foggia (max LAI: 6–7, [37]) and maize in Veneto (values > 6, [38,39]). On maize, the ensemble of the LAI profiles obtained combining crop- and biome-specific equations was the one with the highest number of functions (n = 36), but at the same time showing the lowest variability from April to October, considering the 25–75th and the 5–95th percentile. This means that all the functions built for maize in the Temperate region tend to behave similarly and are consistent in the derived LAI values, despite that they were developed using data from other sensors than Sentinel 2. To the contrary, selecting a combination of crop- and spatial resolution-specific equations reduces the number of available equations (n = 10), constraining the choice to a few alternatives, developed in different biomes. As for the wheat plot, the largest number of functions was recorded in the crop- and spatial resolution-specific equations combination (n = 18). However, the ensemble of functions leading to the lowest LAI variability from February to May, at least for the 25–75th percentile, was derived using the spatial resolution- and biome-specific combination which has few functions (n = 5), but similar. This means that choosing few functions developed in the Mediterranean biome and using Sentinel 2-like sensors, even if for different crops, ensured a higher consistency than relying on wheat-specific equations, in terms of both the LAI values and seasonal trend.

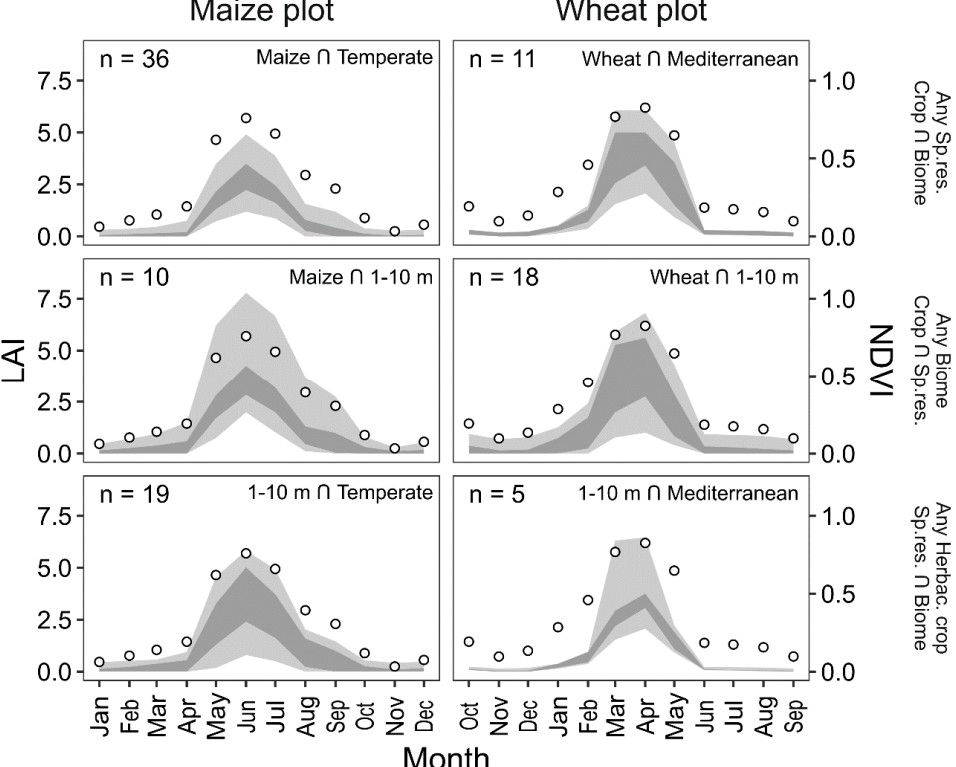

**Figure 5.** Leaf area index (LAI) seasonal profile of the case studies, computed from normalized difference vegetation index (NDVI, dots) values according to the equations derived from the following combinations of attributes: (**top**) crop- and biome-specific equations; (**middle**) crop- and spatial resolution-specific equations; (**bottom**) biome- and spatial resolution-specific equations. The grey shadows indicate the 5–95th (light grey) and the 25–75th percentile (dark grey) of the LAI values obtained from the equations considered.

## 4. Discussion

The key importance of the LAI as a biophysical parameter to characterize crop growth and development is widely recognized and largely explored across disciplines [40–42]. In remote sensing research, a major concern is the quantitative assessment of the LAI [8,11]; currently, there are both generic solutions based on proprietary software algorithms (such as ESA-SNAP) or ready-to-use data provided as satellite products (such as MODIS), and crop ad hoc approaches, developed by means of field measurements, spectral indices, and empirical models. One added value of our contribution with respect to the already available generic LAI products lies exactly in the efforts made by authors of the articles we derived the equations from. Using generic LAI products flattens all the possible variability in the LAI values associated with different crop types, different bioclimatic environments, and the different sensors used. The reason why so many authors did not rely on global LAI products but struggled to develop their own NDVI-LAI conversion equations is the need to have realistic data according to their own cropping systems. With this work, we tried to organize and explore this complexity, adding further value to that provided by the involved scientific literature. In this perspective, the characterization of the different NDVI-LAI equations may enable scientists to also monitor, model, and map the crop growth process, going beyond the field extent and considering wider area coverage, by using existing and upcoming satellite missions, and according to their own research purpose.

A deeper understanding of the state-of-art of the use of vegetation indices, like the NDVI, to estimate the crop LAI is required to grasp the limits and potentialities of the methodologies proposed so far. This work contributed to fill this research gap by reviewing the use of the NDVI to derive the LAI and releasing a RESTful API to make a compendium of (i) the NDVI-LAI equations available in the recent literature, (ii) their characterization, and (iii) their expert-based selection.

The characterization of the NDVI-LAI equations highlighted two main constraints when developing a function: the environmental conditions and the availability of recurring data during the growing season. The heterogeneity of environments in terms of resource availability represents a key discriminant in the development of an NDVI-LAI equation. According to our results, the main differences among functions emerged in biomes with the highest variability in terms of water availability (i.e., biomes which include dry to wet environments); here, the available equations tend to each cover a specific range of bioclimatic conditions, and therefore show a distinct behaviour. This is mainly due to the crop-specific demand of resources, such as water and light, that is not only the expression of the crop genotype but also of the management and weather conditions [43,44]. The latter determine the form and behaviour of the NDVI-LAI function, directly by affecting the crop cycle (e.g., anticipating or delaying the achievement of phenological phases, shortening or lengthening the growing season), through precipitation and temperatures, and indirectly by influencing the availability of reliable, not-cloudy satellite data.

Having at one's disposal a dense time-series of NDVI data allows to develop a more realistic NDVI-LAI equation, able to represent the whole crop season and take into consideration the different crop phenological phases. In this sense, high temporal resolution satellites, like MODIS, though providing low spatial resolution data, are especially suitable for crop LAI estimation over large areas [45]. Furthermore, unlike high-resolution satellite missions such as Sentinel 2, MODIS operationally produces temporally aggregated images (i.e., composites) which consider the most reliable observations within a time window and therefore are little affected by cloud cover [10]. Our results confirmed a higher consistency in the LAI derivation with equations using low rather than very high spatial resolution satellites on both crops, the former ensuring good quality data at a regular time interval throughout the growing season. Accordingly, when only a few spot data are available, the linear equations are commonly used, reflecting single moments during the season rather than the whole growing process of the crop. To the contrary, when a large number of NDVI seasonal observations are available, the exponential relationship is mostly used, being capable of reflecting non-linearities and the NDVI saturation at canopy close stages. Our

results showed how, regardless of crops, biomes, and sensors, the NDVI-LAI relationship tends to plateau around an LAI ≈ 5. This evidence has been confirmed by other authors, e.g., [8,26] underlined that as the LAI exceeds 2, the NDVI is generally insensitive to detect LAI changes in grasses, cereals, and broadleaf crops; [46,47] found an exponential relationship between the NDVI and LAI and highlighted that the NDVI is not sensitive at LAI > 3 on wheat, soybean, and corn. Such evidence underlines that the choice of the mathematical form, e.g., exponential, power, or logarithmic, requires meaningful parameters (e.g., in terms of saturation) in order to reliably reflect the crop growth process and phenology.

The physical and biological properties of a vegetation canopy vary along with the phenological development, affecting the seasonal profiles of the NDVI, and in turn of the LAI [2]. When considered, the phenological component was generally taken into account, developing as many conversion functions as the different phenological events (i.e., tillering, elongation, flowering, grain-filling, and maturity; e.g., [48]). However, even though the crop phenological evolution is essential to characterize the LAI-NDVI relationship, it has been largely neglected in the available studies, which mostly use a single regression equation to only reproduce the period of "green" and active vegetation, without considering the early beginning and the end of the growing season [2]. On the other hand, data availability is an unavoidable constraint that necessarily determines methodological choices; as observed by [49] while developing a physically based algorithm for the estimation of the LAI from NDVI observations, where "the algorithm must be viewed within a framework dominated largely by practical consideration, and to a lesser extent by accuracy".

## 5. Conclusions

Our study demonstrated that the choice of the most suitable NDVI-LAI equation depends on data accessibility, the scale of the investigation, and the location of the study area, and therefore a trade-off of priorities is needed. In this sense, this study underlined that there is no need for preferring one single equation with respect to an ensemble of different equations and that knowing the variability of the LAI estimations allows for associating a degree of reliability/uncertainty to the specific approach: rather than being a question of estimation accuracy, it is a problem of output consistency. To the best of our knowledge, this is something never done before and with useful implications for scientists who cannot rely on field-observed LAI data for different reasons (e.g., a lack of resources, retrieval difficulties, a large-scale approach, high temporal resolution requirements, etc.).

This work contributed to the development of a RESTful API to foster the choice of the most suitable NDVI-LAI equations for further studies. The main strengths of this API are (i) the practicality of collecting a battery of equations in a single component, (ii) the possibility of comparing the outputs of different equations with the same input NDVI dataset, and (iii) the opportunity for checking the unrealistic outputs when the equations are applied on new case studies. As for technological aspects, software architecture allows an ease of maintenance and ensures extensibility, concerning both new equations and input bands. The release of an SaaS product solves long-standing issues related to the installation, configuration, and updating of the software, transferring problems connected to computing capacity to the service provider. Placed between satellite data sources and crop models, the SaaS could enable performing large-scale simulations, overcoming issues related to the application limits of the single functions by relying on the "wisdom of the crowd" of the function ensemble. Furthermore, the software service is ultimately global and available, and it can be crowdsourced to users all over the world for verification, correction, and improvement to expand the usability of the service.

Exploiting the potentialities of artificial intelligence methods for interpreting remote sensing data and estimating vegetation characteristics, in terms of high computational efficiency and the ability to accurately approximate complex non-linear functions, could represent an alternative way to enrich the system and extract information from multi-dimensional data, without subjective effects.

**Supplementary Materials:** The following supporting information can be downloaded at: https://www.mdpi.com/article/10.3390/rs14153554/s1, File S1: NDVI-LAI equation library: API call documentation.

**Author Contributions:** Conceptualization, S.B. and S.U.M.B.; methodology, F.G. and F.S.; formal analysis, F.G. and F.S.; software development, F.G., F.S., M.S. and D.F.; data curation, D.M., E.R. and M.S.; writing—original draft preparation, S.B.; writing—review and editing, S.B., S.U.M.B., F.G., F.S. and D.F. All authors have read and agreed to the published version of the manuscript.

**Funding:** This research was funded by the Italian Ministry of Agriculture, AgriDigit program (DM 36503.7305.2018 of 20 December 2018).

**Data Availability Statement:** Not applicable.

**Conflicts of Interest:** The authors declare no conflict of interest. The funders had no role in the design of the study; in the collection, analyses, or interpretation of data; in the writing of the manuscript; or in the decision to publish the results.

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
