# Peer review of "On the Use of NDVI to Estimate LAI in Field Crops: Implementing a Conversion Equation Library"

_remotesensing, doi:10.3390/rs14153554_

Round 1

Reviewer 1 Report

This manuscript is valuable for readers of "Romote Sensing" journal. I could not find any problems for its publication.

Author Response

R#1

This manuscript is valuable for readers of "Remote Sensing" journal. I could not find any problems for its publication.

> We thank the referee for the positive comments.

Reviewer 2 Report

The article investigates a multitude of available research articles upon the relation of leaf area index and normalized difference vegetation index and the underlaying equations of the proposed models while taking into account the environmental, temporal and biome factors within the studies. Furthermore, a software solution for the analysis has been developed. The results are well summarized and of interest for scientists working within the field of plant based remote sensing.

The article is well written and structured. The additions which have been made within the article are helpful to elucidate the article’s importance to readers.

Specific comments are as follows:

At multiple occasions the authors refer to research articles as papers, this should be corrected to avoid slang.

Line 48: Change “Remote sensing has then widely used to” to “Remote sensing has then widely been used to”.

Line 190 – 192: Please correct the grammar of the sentence.

Line 292: Change 1-10m and 10-30m into 1-10 m and 10-30 m respectively to conform with the style for SI units used within the article.

Figure 1: The figure legend uses (a), (b) and (c), which are not present in the figure. Either add at appropriate places or replace with top, middle and bottom images.

Author Response

R#2

The article investigates a multitude of available research articles upon the relation of leaf area index and normalized difference vegetation index and the underlaying equations of the proposed models while taking into account the environmental, temporal and biome factors within the studies. Furthermore, a software solution for the analysis has been developed. The results are well summarized and of interest for scientists working within the field of plant based remote sensing.

The article is well written and structured. The additions which have been made within the article are helpful to elucidate the article’s importance to readers.

> We thank the referee for the positive comments.

Specific comments are as follows:

At multiple occasions the authors refer to research articles as papers, this should be corrected to avoid slang.

> We followed the referee’s suggestion and change the term “paper” with “article” or “work” across the whole manuscript.

Line 48: Change “Remote sensing has then widely used to” to “Remote sensing has then widely been used to”.

> Done.

Line 190 – 192: Please correct the grammar of the sentence.

> We changed the sentence as follows: “Then, we referred to the freely available Sentinel 2 satellite data because of the high spatial resolution, to minimize the probability of having mixed pixels”.

Line 292: Change 1-10m and 10-30m into 1-10 m and 10-30 m respectively to conform with the style for SI units used within the article.

> Done.

Figure 1: The figure legend uses (a), (b) and (c), which are not present in the figure. Either add at appropriate places or replace with top, middle and bottom images.

> Done.

Reviewer 3 Report

This presents the use of NDVI to estimate LAI for field crop mapping. The article is in good shape and could be considered for possible publication after addressing the comments given below:

Title: OK. If the focus is the field crops then the methods are also indicating forest and mixed land covers.

Abstract: Sounds good.

Line 11 “vegetation fitness” may not be an appropriate term.

Line 19 The criteria for the literature research should be described briefly

Materials and Methods

Methods are technically strong and well explained.

It is important to specify the research focus area, which is probably agriculture/crop.

Authors should also note and discuss the saturation of the NDVI with a dense canopy and the relationship LAI-NDVI might mislead in such cases.

Line 103: Should be included in the abstract.

Line 106-07:  The scope of the work is very broad, NDVI of forest and mixed land cover could behave differently, in addition, the seasonality and physical structure of various crops might end up differently. How the NDVI-LAI relation varies with the crop development phases.

Results:

Results are described in detail.

Figure 4: Does the NDVI is indicating saturation in most of the cases?

Figure 5: It would be good to plot the NDVI bases seasonality also, on the secondary y-axis, to understand the differences and effectiveness of the corresponding index, and to see which phenology is better.

Discussion

It would be good to add relevance and perspective to the current study for wider area coverage using existing and upcoming satellite missions.

Conclusion

No Comments. 

Author Response

R#3

This presents the use of NDVI to estimate LAI for field crop mapping. The article is in good shape and could be considered for possible publication after addressing the comments given below:

Title: OK. If the focus is the field crops then the methods are also indicating forest and mixed land covers.

> We removed the forest covers from the data. We kept mixed land covers because they can include field crops.

Abstract: Sounds good.

> Thank you.

Line 11 “vegetation fitness” may not be an appropriate term.

> We changed “fitness” into “activity”.

Line 19 The criteria for the literature research should be described briefly

> We see the referee’s point and added this line in the Abstract: “We conducted a literature search using “NDVI” AND “LAI” AND “crop” as search string focusing on the period 2017-2021”.

Materials and Methods

Methods are technically strong and well explained.

> Thank you.

It is important to specify the research focus area, which is probably agriculture/crop.

> We see the referee’s point and modified accordingly the first line of paragraph 2.1 as follows: “We conducted a literature search on agriculture and crops using Google Scholar database…”.

Authors should also note and discuss the saturation of the NDVI with a dense canopy and the relationship LAI-NDVI might mislead in such cases.

> We noted and faced the issue of saturation in many different parts of the paper: L73, L224, L263, L294, L392, L400. However, according to the referee’s comment, in the Methods section we added this line (L131-132): “As for NDVI saturation, when LAI values were not upper limited…”.

Line 103: Should be included in the abstract.

> Done.

Line 106-07:  The scope of the work is very broad, NDVI of forest and mixed land cover could behave differently,

> As replied above, we removed the forest covers from the data. We kept mixed land covers because they can include field crops.

in addition, the seasonality and physical structure of various crops might end up differently. How the NDVI-LAI relation varies with the crop development phases.

>The referee is right. However, he/she has to consider that we are only collecting different equations sorting them per crops, sensors and biomes, we are not providing our own algorithm; so, although interesting, we do not have the tools to investigate the phenological variability of the NDVI-LAI equations. Furthermore, please consider that only very few papers, among those retrieved, provided different equations for the same experimental case study according to the different phenological phases (L402-410).

Results:

Results are described in detail.

> Thank you.

Figure 4: Does the NDVI is indicating saturation in most of the cases?

> Yes, and we reported it in the corresponding part of the Results section (see L292-295).

Figure 5: It would be good to plot the NDVI bases seasonality also, on the secondary y-axis, to understand the differences and effectiveness of the corresponding index, and to see which phenology is better.

> We thank the referee for the useful comment. We added the NDVI profile as suggested.

Discussion

It would be good to add relevance and perspective to the current study for wider area coverage using existing and upcoming satellite missions.

> We followed the referee’s suggestion and wrote the following lines (L354-357): “In this perspective, the characterization of the different NDVI-LAI equations may enable scientists also to monitor, model and map the crop growth process going beyond the field extent, and considering wider area coverage, by using existing and upcoming satellite missions, and according to their own research purpose.”

Conclusion

No Comments.

> Thank you.